# Altared Environments:
# The Role of Normative Infrastructure in AI Alignment

## Abstract

Cooperation is central to human life, distinguishing humans as ultra-cooperative among mammals. We form stable groups that enhance welfare through mutual protection, knowledge sharing, and economic exchanges. As artificial intelligence gains autonomy in shared environments, ensuring AI agents can engage in cooperative behaviors is crucial. Research in AI views this as an alignment challenge and frames it in terms of embedding norms and values in AI systems. Such an approach, while promising, neglects how humans achieve stable cooperation through *normative infrastructure*. This infrastructure establishes shared norms enforced by agents who recognize and sanction norm violations. Using multi-agent reinforcement learning (MARL), we investigate the impact of normative infrastructure on agents' learning dynamics and their cooperative abilities in mixed-motive games. We introduce the concept of an ***altar***, an environmental feature that encodes actions deemed sanctionable by a group of agents. Comparing the performance of simple, independent learning agents in environments with and without the altar, we assess the potential of normative infrastructure in facilitating AI agent alignment to foster stable cooperation.

## 1. Introduction

The Alignment challenge – how do we get individuals to choose behaviors that benefit the groups they live in – has been around as long as humans have. In the context of humans, this pertains to the question of how we have achieved such ultra-cooperative societies, far beyond anything we see in other mammals. In AI research, the alignment challenge has largely been framed in terms of how we embed

[1]Anonymous Institution, Anonymous City, Anonymous Region, Anonymous Country. Correspondence to: Anonymous Author <anon.email@domain.com>.

Preliminary work. Under review by the International Conference on Machine Learning (ICML). Do not distribute.

values and norms into AI systems. But this approach is inherently limited: it is not possible to articulate all our values and norms (Hadfield-Menell & Hadfield, 2018); norms and values are dynamic, constantly adapting to changes in environments/populations/information; and values and norms are highly contested. Further, AI developers who embed values (e.g. RLHF, Constitutional AI (Bai et al., 2022)) lack the legitimacy to choose these norms.

Most importantly, this is not how humans solve the alignment (cooperation) challenge: they do not encode specific values and norms–in fact, they are highly adaptable to changes in values and norms. Humans solve the problem with institutions that articulate and enforce a set of (variable) rules. Humans are *normatively competent* in the sense that they possess the cognitive architecture and learned behaviors that allow them to process, respond to, and help constitute normative infrastructure and normative social orders. Norms and values are not data, as approaches such as RLHF assume; they are the equilibrium outcomes of complex dynamic normative systems. We argue in this paper that alignment research should shift its focus from *how to embed values in AI agents → how to make AI agents normatively competent*. (Perolat et al., 2017) can be seen as an early contribution on this research agenda, endowing agents with a punishment (sanctioning) technology which allowed agents to find solutions to common pool problems. (Köster et al., 2022) took this a step further by implementing a hidden classification institution that rewarded agents for sanctioning behaviors that reduced individual welfare.

(Hadfield & Weingast, 2012) propose a parsimonious rational agent model of normative social order, consisting of a *classification institution* that provides common knowledge binary classification of all behaviors as either "punishable" or "not punishable" (possibly through complex application of general principles to particular cases) and an *enforcement mechanism* that incentivizes agents to prefer "not punishable" actions. A stable normative social order is achieved when most agents are mostly in compliance and avoiding punishment. (Hadfield & Weingast, 2012) focus in particular on the case, which describes most of human history and much of modern life as well, in which punishment is primarily delivered by ordinary agents (rather than spe-

cialized enforcers (Hadfield & Weingast, 2013).) Agents must therefore be incentivized and coordinated to engage in costly third-party punishment (which could be relatively mild, such as criticism, or more harsh, such as exclusion from the group) and to condition such punishment actions on a shared classification institution. Although shared classification could be entirely emergent and informal, groups that converge on a single authoritative (more formal) classification institution–such as a chief, a group of elders, or a court–can enjoy the benefits of maintaining cooperation even in the face of changing environments and populations (Hadfield, 2017). Inspired from this line of literature, in this paper, we introduce a classification institution as an observable feature of the environment, which we call an *"altar"* (appealing to the idea that humans developed sacred places that represented authoritative sources for rules). Using multi-agent reinforcement learning, we investigate the implications of such normative infrastructure on the learning dynamics and alignment capabilities of the agents interacting in mixed motive games. We demonstrate the value of this approach by showing that agents in our altar environment outperform agents in a hidden rules environment (and both outperform agents in an environment without rules) at achieving cooperation in the Allelopathic Harvest game which presents the challenge of alignment, coordination and free-riding for the agents interacting in the environment.

## 2. Related Work

The problem of cooperation, that is, how to design environments and algorithms to align learning agents' behaviour towards higher collective welfare, has seen an increasing focus in multiagent literature (Du et al., 2023). Drawing from how human societies have solved the cooperation problem (Boyd & Richerson, 1992), mechanisms such as third-party punishment have shown promise as an approach in multiagent reinforcement learning based artificial agents (Köster et al., 2022). However, most techniques focus on the emergence of cooperation by adapting the agents' reward function to include individual behavioural incentives that can eventually assist in the convergence to collective-welfare optimizing behaviour. These incentives include making agents more altruistic (McKee et al., 2020), incentives that make agents mimic each others' punishment behaviour (Vinitsky et al., 2023), care about the reputation of the agent in the population (McKee et al., 2021), direct punishment (Dasgupta & Musolesi, 2023), and social-learning mechanisms that enables agent to learn from experts(Ndousse et al., 2021).

In contrast to this above set of techniques, our method follows the key insight that human societies did not learn to be cooperative just through exploration and individual behaviour change. Rather, cooperation follows as a second-order effect once societies learn to coordinate their peer

sanctions through social structures, such as informal norms and formal institutions (Richerson & Boyd, 2008; Henrich, 2016). Our work focuses on a particular manifestation of these structures, namely, classification institutions, that announce right and wrong behaviours around which agents can voluntarily coordinate their sanctioning behaviour (Hadfield & Weingast, 2012). More importantly, compared to previous work in MARL, we shift the focus from individual learning to learning about social structures. Specifically, our work uses standard MARL methods to give agents the ability to recognize features of authoritative classification institutions (the *altars*) that represent the norms of a population.

## 3. Preliminaries

A **Markov Game**, is a generalization of a Markov Decision Process (MDP) to a multi-agent setting and is formally defined as follows. There is a set of states $S$, and each agent $i$ has a set of actions $A_i$. The transition function $T : S \times A_1 \times A_2 \times \cdots \times A_N \to \Delta(S)$ determines the probability distribution over states, where $\Delta(S)$ denotes the probability distribution over states. Each agent $i$ has a reward function $R_i : S \times A_1 \times A_2 \times \cdots \times A_N \to \mathbb{R}$. Additionally, there is a discount factor $\gamma \in [0, 1)$. In each state $s \in S$, each agent $i$ selects an action $a_i \in A_i$. The next state $s'$ is determined by the transition function $T$, and each agent $i$ receives a reward $R_i$. A **Partially Observable Markov Game (POMG)** extends the concept of a Markov Game to scenarios where agents have limited observations of the state. In addition to the components defined for a Markov Game, each agent $i$ in a POMG has a set of observations $O_i$. The observation function $O : S \times A_i \to \Delta(O_i)$ determines the probability distribution over observations. In a POMG, each agent $i$ selects an action $a_i \in A_i$ based on its observation $o_i \in O_i$, rather than the full state $s$. The next state $s'$ is determined by the transition function $T$, and each agent $i$ receives a reward $R_i$. A **Markov game with sanctions** allows one of the actions $a_i \in A_i$ for each agent to be the use of a zapping beam to sanction another agent. This action imposes a negative consequence (e.g., a penalty or loss of reward) on the targeted agent.

**Multi-Agent Reinforcement Learning** (MARL) involves multiple agents interacting within a shared environment, aiming to optimize their individual or collective behaviors. MARL is often applied to solve Markov games, where the objective is for each agent to learn a policy that maximizes its expected cumulative reward. Formally, given a state $s \in S$, each agent $i \in N$ chooses an action $u_i$ and obtains a reward $r(s, u)$ with a private observation $o_i \in O_i$, where $u = \{u_i\}_{i=1}^N$ is the joint action. The joint policy of the agents is denoted as $\pi_\theta = \{\pi_\theta^i\}_{i=1}^N$ where $\pi_\theta^i : S \times A_i \to [0, 1]$ is the policy for agent $i$. The objective of each agent is to maximize its total expected return $R_i = \sum_{t=0}^{\infty} \gamma^t r_t^i$.

## 4. The Altar Approach

In this work, we employ a computational approach to investigate the implications of the presence of an **"altar"**, a normative institution that encodes the representation of the norm content, as a feature of an environment in which self-interested reinforcement learning agents need to coordinate effectively to maximize social welfare. We describe a mechanism through which the altar can benefit a society. Our argument is based on the dynamics of learning in a group that lacks a priori knowledge of the important aspects of their social order. The altar serves as an explicit feature of the environment, providing a clear classification of what is approved versus disapproved within the given system.

As a starting point, we consider the setup in (Köster et al., 2022), where agents can sanction each other to enforce a rule and the environment employs a hidden classification rule via reward signals that guide the sanctioning behavior of agents. We extend this setup to include an explicit altar feature in the environment that contains the normative information associated with the reward signals; at the start of training this association between the information on the altar and the reward signals for punishment is unknown to the agents. Our environment is inspired by the *Allelopathic Harvest* game (Agapiou et al., 2023; Köster et al., 2020), which poses both the coordination and the free-rider problem, making it challenging for agents to reach a welfare maximizing outcome. Specifically, in our environment, called the **Altared Allelopathic Harvest**, there are berries of three different colors and sixteen agents can plant and consume berries. Agents get reward for consuming any colored berry (+1) but receive higher reward for consuming their preferred color berry (+2). Planting does not generate any reward or cost and hence agents have no direct incentive to plant, leading to a free-rider problem. The agents can only consume ripened berries and the berry ripening rate is directly proportional to the fraction of the largest amount of berry color. Hence, if all three colors are equally distributed, berries will have the slowest ripening rate and achieving a monoculture of a single berry color will generate the highest berry ripening rate, thereby giving a chance to agents to accumulate more reward. This is a coordination problem.

Agents also have a zapping action that they can use to sanction other agents. When an agent (called a target agent) is sanctioned, it receives a penalty of -10. Similar to the hidden classification rule in (Köster et al., 2022), the sanctioning agent (called a source agent) incurs a cost of -10. However, with a hidden classification rule, a desired planting behavior (berry color) is established at the start of game and planting this color is considered the correct thing to do. Hence, if the source agent sanctions a target for planting any color other than correct one, or failing to plant berries (free-riding), it receives a reward of +20 (leading to net positive of +10).

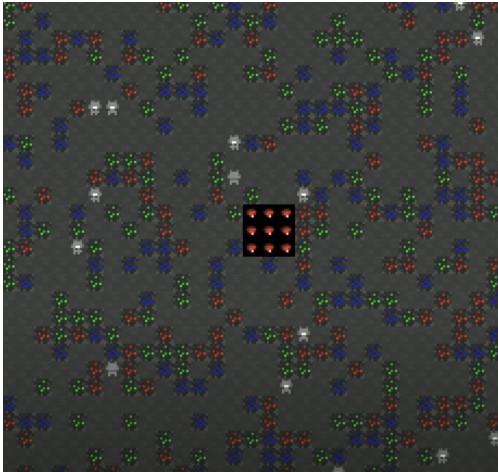

*Figure 1.* Altared version of Allelopathic harvest. Altar is represented as a 3x3 observation in the center with the color of berry on it for which a monoculture is desired.

We modify this approach by introducing the altar in the environment – a visual observation in the center of the map that displays the approved berry color. Figure 1 provides an example with an altar that declares red berries to be the community's desired monoculture. While unknown to the agents, the rewards for punishment match this norm content - that is agents face the task of learning that they should punish agents who do not plant red berries. Agents become marked with the color of the berry they have most recently planted and become grey with some probability after eating until they plant, thus allowing other agents to identify agents who have not followed the planting rule on the altar.

We posit that the presence of the altar allows agents to more quickly understand the normative social order established in their community. At the start of the training, the agents will receive normative guidance via reward signals for punishment as in (Köster et al., 2022), however, as soon as the agent steps on the "altar", it will now have the content of that altar as a part of its observation space. The prediction is that the agents will learn to correlate the information on the altar with the reward they receive for sanctioning correctly and the cost they incur for sanctioning incorrectly. We expect the learning to progress in three phases: (i) agents learn to recognise the norm content on the altar and correlate it with the sanctioning reward/penalty; (ii) agents learn to correctly enforce punishment for planting the wrong color berries; and (iii) agents learn to comply with the norm via third party enforcement received in the forms of sanctions for planting wrong colored berries. If learning is successful, free-riding is limited and a monoculture is achieved.

In this work, we test the hypothesis that by providing the altar as a piece of normative infrastructure (a constructed feature in the environment) that serves as an authoritative

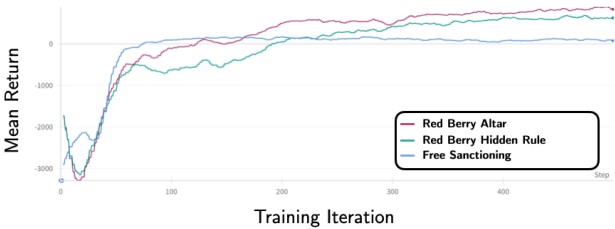

*Figure 2.* Rewards of training agents. Please note that we have adjusted the reward curves to only include reward for berry consumption and penalty for getting sanctioned while removing any effect of reward and cost received due to sanctioning rules

reference for a community's rules, agents can achieve faster coordination in a less costly manner than they can when rules are hidden and third-party enforcement and compliance behaviors are wholly emergent.

## 5. Experiments

In this section, we provide details about various baseline conditions and training procedure and conclude with discussion on empirical results. For the experiments in this section, out of 16 agents, 8 preferred red berries and other 8 preferred green berries and the altar displayed red colored berry, making red monoculture the desirable outcome.

### 5.1. Baseline and Altar Conditions

In our experiments, we tested three distinct environment conditions to explore the effects of different sanctioning mechanisms on agent's enforcement and compliance behavior and their ability to achieve a monoculture:

**Free Sanctioning.** In this condition, there is no altar or hidden rule in the environment. Agents can freely zap other agents, with the target agent receiving a penalty of -10 points. The source agent, however, does not receive any reward or penalty for zapping.

**Hidden Rule (Red).** In this condition, there is no visible altar in the environment, but there is a "hidden rule." If a source agent zaps another agent that is red, both the source and target agents receive a penalty of -10 points. If the source agent zaps an agent of any other color (green, blue, white), the source agent receives a net reward of +10 points.

**Altar (Red).** This condition features a 3x3 altar in the center of the map displaying red berries (see Figure 1). Agents have an augmented observation space that includes a memory slot, which starts as empty. When an agent enters a tile that is part of the altar, their memory slot updates to altar observation. If a source agent zaps a target agent of the same color displayed on the altar (red), both the source and target agents receive a penalty of -10 points. If the source agent

zaps a target of any other color, the source agent receives a net reward of +10 points, while the target still receives a penalty of -10 points.

### 5.2. Training Details

We utilized the Proximal Policy Optimization (PPO) algorithm to train 16 independent learning agents. The training setup employed a single GPU, with a training batch size of 32,000, an SGD minibatch size of 16,000, and a rollout length of 100. Each episode consisted of 2000 steps. The agent architecture consisted of fully connected layers with hidden sizes of 64 and 256, using ReLU activations.

### 5.3. Results

As displayed in Figure 2, we first examine the performance of the agents towards achieving high welfare, measured as sum of rewards across agents and averaged over episodes. In this environment, agent can maximize their welfare by aligning their planting and sanctioning behavior so as to achieve a monoculture of any one berry color. As berry ripening rate is proportional to the largest fraction of berry color, a monoculture would lead to fastest ripening of berries, thereby allowing agents to accumulate maximum possible reward over time. Our experiments demonstrate that the agents learning in the condition with an "altar" in the environment achieves highest reward and also consistently tracks higher reward across training compared to the hidden rules baseline. We conjecture that the availability of the altar observation helps the agents to learn to correlate the norm content to the correct enforcement behavior and help them align their sanctioning behavior in a manner that is beneficial for learning second order compliance behavior. The altar information serves as a tool that agents can leverage to enhance their learning of coordination behavior, reducing the necessary training time and effectively decreasing the cost of achieving coordination. Further, it is hard for agents to learn compliance behaviors in the free sanctioning condition without any signal to learn any meaningful enforcement behavior.

To understand how reward performance is reflected in the monoculture fraction achieved by each group of agents and the time taken under different settings, we visualize the berry map midway of an episode (total length 2000 steps), for each condition in Figure 5. For the free sanctioning condition, the agents fail to align over a single berry color and therefore are unable to increase the fraction of any one berry. Agents are able to move towards a monoculture and have achieved over 95% monoculture in both hidden rule condition and in the Altared environment. However, one can observe that the agents are more aligned in the Altared environment on planting red behaviors (see red-colored agents) while in the hidden rule condition, there are still several free-riding agents or agents that plant some other berry color.

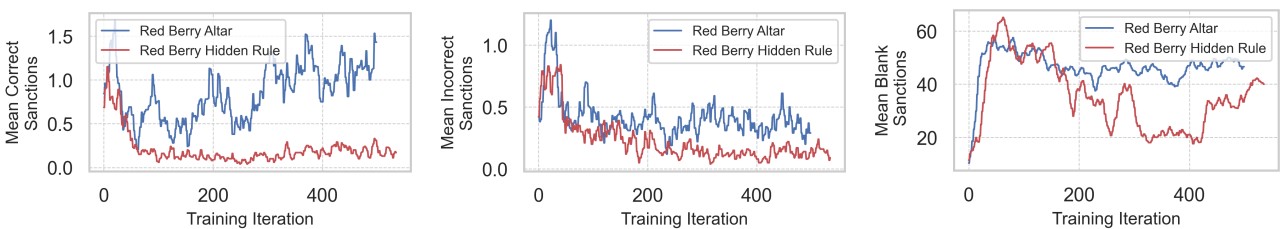

(a) Correct sanctions vs Training iterations    (b) Incorrect sanctions vs Training iterations    (c) Blank zaps vs Training iterations

*Figure 3.* Sanctioning Behavior of agents across training period

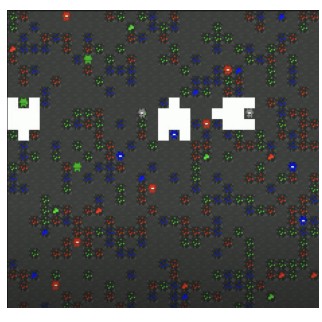
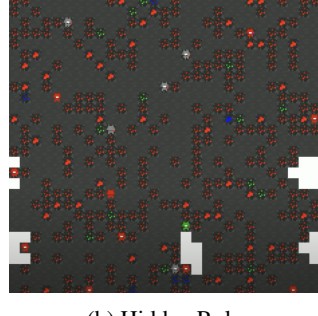
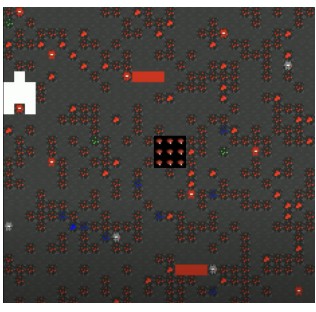

(a) Free Sanctions                                (b) Hidden Rule                                    (c) Altared

*Figure 4.* Monoculture and Agents' status at halfway of an episode for trained agents

Finally, we assess the sanctioning behavior of the agents and their ability to use the reward signal and classification institution (altar) to learn to sanction the disapproved actions correctly. Figure 3 demonstrates that for both hidden rule and altared environment, the incorrect sanctioning behavior decreases over time signifying that agents are able to avoid wrongly punishing other agents because they want to avoid second-order punishment from the environment (wrong sanctioning is costly). The more interesting result is that in the altared environment, agents show increasing number of correct sanctions across training. This is important as there is a chance of agents free-riding and forgetting to plant berries once the monoculture is achieved. Sanctioning any free-riding agent is the correct thing to do and agents learn to enforce this in altared environment while they stop sanctioning in general in the hidden rule environment. Finally, our setup allows for random zapping in the blank area and agents do take this action quite frequently as it is not costly. To discern the ability of agents to learn correct enforcement behavior, we report these numbers separately in Figure 3(c). In Appendix A, we provide more results depicting the fraction of monoculture achieved by both the hidden rule and Altar approaches at various time during training.

## 6. Concluding Remarks

In this work, we address the alignment challenge by proposing a shift in focus from embedding values in AI agents to making them normatively competent. To advance this direction, we introduce the ***altar***, a classification institution as a feature of a multi-agent environment that aids in solving coordination problems. Using multi-agent reinforcement learning, we examine the impact of this infrastructure on learning dynamics, enforcement, and compliance behaviors. In a modified Allelopathic Harvest game, we demonstrate superior performance of agents trained with an altar compared to those without it. While our experiments use a simple setup to demonstrate the altar's effectiveness in addressing the alignment challenge, our objective is to lay the groundwork for future research in AI alignment and multi-agent cooperation. First, our setup features an institution with fixed information, but future research could explore institutions with norm content that changes across or within episodes, as well as multiple institutions with conflicting, unreliable, or converging information. Next, we posit that our approach will be particularly effective in promoting generalization, adaptability, and robustness across environments with different normative institutions. Agents that learn to recognize altar information and correlate enforcement behaviors with norm content will adapt and train quickly when transferred to new environments. Finally, this work has significant implications for MARL environment design research. As the field advances towards designing environments that capture diverse social and economic phenomena, we suggest that incorporating normative infrastructures should be a priority in developing the next generation of MARL environments.

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

## A. Progress towards achieving monoculture over training

Figure 5 demonstrates the progress of agents coordination behavior and consequent monoculture achieved by the agents across training period. The top row demonstrates the agents training in Altared environment while the bottom row shows the agents training in hidden rule setup. As one can observe, agents training in Alterad environment consistetly show higher monoculture fraction at any given training point which tracks with the higher welfare it gets across the training period. Further, the agents trained in Altared environment show better alignment with the normative order as most agents have started stopped planting non-red berries or free-riding by iteration 400 which is not true for the hidden rule setup. This provides insight into the ability of the altared environment to support cost effective coordination among agents, eventually resulting in the higher social welfare.

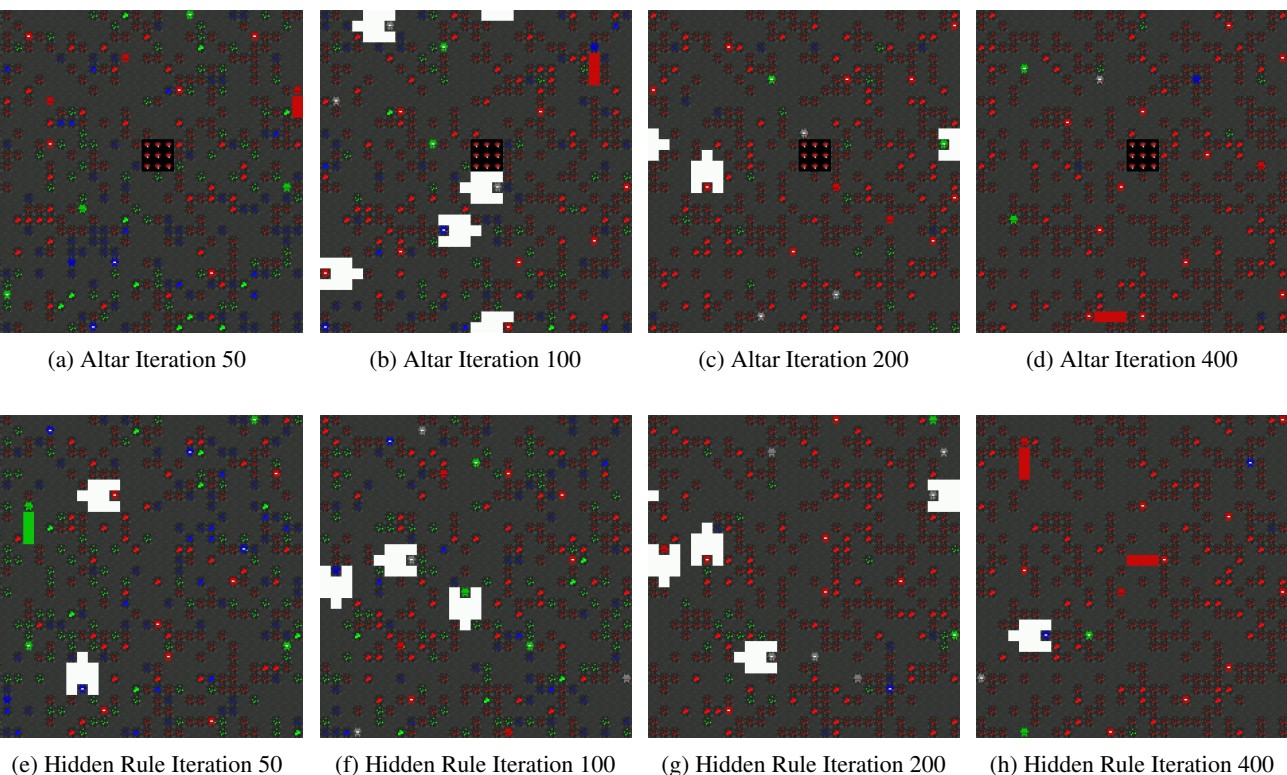

| (a) Altar Iteration 50 | (b) Altar Iteration 100 | (c) Altar Iteration 200 | (d) Altar Iteration 400 |

| (e) Hidden Rule Iteration 50 | (f) Hidden Rule Iteration 100 | (g) Hidden Rule Iteration 200 | (h) Hidden Rule Iteration 400 |

*Figure 5.* Monoculture and Agents' status at halfway of an episode for trained agents

