# OpenReview forum: "Altared Environments: The Role of Normative Infrastructure in AI Alignment"
_ICML.cc/2024/Workshop/Agentic_Markets — Agentic Markets @ ICML'24 Poster_

### Official Review · Reviewer_G2hc · 2024-06-12
**Review for "Altared Environments: The Role of Normative Infrastructure in AI Alignment"**

**Rating:** 8
**Confidence:** 3

**Review:**

1) Description -
Integrating moral values and norms into AI agents is a challenging task because humans often disagree on these matters due to their subjective and dynamic nature. Instead of focusing on how to incorporate norms into AI, the authors suggest a shift towards enabling AI agents to process normative social structures similar to humans. To achieve this, they propose employing a shared classification mechanism as the foundation for value judgments (including punishments and rewards). By utilizing multi-agent reinforcement learning (MARL), the authors explore the learning dynamics resulting from this classification institution.

2) Strengths
- This workshop paper is very well-written and presents a clear, easy-to-follow structure. It effectively articulates the challenges associated with the proposed approach.
- The initial results demonstrate the significant potential of this recommended method, as it offers cost-effective coordination capabilities.

3) Weaknesses
- This method may encounter analogous issues to those faced by human decision-making systems. Specifically, the absence of a universally accepted objective institution to govern norms can lead to problems, as different institutions (e.g., those separated geographically) might have conflicting normative values. So this will highly depends on who/what is setting up "the altar".

4) Questions/Comments/Enhancements
- Although the paper does not address the scalability aspect of this approach, it would be valuable, perhaps in the future, to examine its practicality in large-scale scenarios, considering the point raised earlier regarding the potential for conflicting norms among diverse institutions.
- Moreover, you could discuss how this approach might be extended to better handle situations where norms and values change over time.
- Additionally, given the promising initial results, I am curious about the subsequent steps planned to further investigate and confirm the efficacy of this suggested approach.